# Classification of Arc-Transitive Elementary Abelian Covers of the *C*13 Graph

**Qianru Xiao** [1,2]*, **Aysha Khan** [3], **Narges Mehdipoor** [4] and **Ali Asghar Talebi** [4]

1   Institute of Computing Science and Technology, Guangzhou University, Guangzhou 510006, China
2   School of Computer Science of Information Technology, Qiannan Normal University for Nationalities, Duyun 558000, China
3   Department of Mathematics, Prince Sattam Bin Abdulaziz University, Al-Kharj 11991, Saudi Arabia; a.aysha@psau.edu.sa
4   Department of Mathematics, University of Mazandaran, Babolsar 4741613534, Iran; nargesmehdipoor@yahoo.com (N.M.); a.talebi@umz.ac.ir (A.A.T.)
*   Correspondence: qianrushaw@gzhu.edu.cn

**Abstract:** Let $\Gamma$ be a graph and $G \leqslant Aut(\Gamma)$. A graph $\Gamma$ can be called *G*-arc-transitive (GAT) if $G$ acts transitively on its arc set. A regular covering projection $p : \overline{\Gamma} \to \Gamma$ is arc-transitive (AT) if an AT subgroup of $Aut(\Gamma)$ lifts under $p$. In this study, by applying a number of concepts in linear algebra such as invariant subspaces (IVs) of matrix groups (MGs), we discuss regular AT elementary abelian covers (R-AT-EA-covers) of the *C*13 graph.

**Keywords:** MGs; IVs; homology group; *C*13 graph; AT graphs; regular covering; lifting automorphisms





## 1. Introduction

All graphs given in this paper are assumed to be finite, connected, and simple. For a graph $\Gamma$, we denote a set of vertices, set of edges, set of arcs, and full automorphism group with $V(\Gamma)$, $E(\Gamma)$, $A(\Gamma)$, and $Aut(\Gamma)$, respectively. Suppose that $G$ is a subgroup of $Aut(\Gamma)$. For $a, b \in V(\Gamma)$, $\{a, b\}$ is the edge located between $a$ and $b$ in $\Gamma$, and $N_\Gamma(a)$ denotes the set of vertices adjacent to $a$ in $\Gamma$ (neighborhood of $a$).

Suppose that $\Gamma$ and $\overline{\Gamma}$ are two graphs. We say that a graph epimorphism $p : \overline{\Gamma} \longrightarrow \Gamma$ is covering projection if $p$ is a local isomorphism, that is, for each $\overline{v} \in V(\overline{\Gamma})$, the restriction $p$ to $N_{\overline{\Gamma}}(\overline{v})$ is a bijection to $N_\Gamma(p(\overline{v}))$, where $p(\overline{v}) \in V(\Gamma)$. We say that $\overline{\Gamma}$ is the covering graph and $\Gamma$ is the base graph. A permutation group $G$ on a set $\Delta$ is said to be semiregular if the stabilizer $G_v$ of $v$ in $G$ is trivial for each $v \in \Delta$. If $G$ is transitive, and semiregular, it is regular. Let $N$ be a subgroup of $Aut(\Gamma)$ such that $N$ is not transitive on $V(\Gamma)$. The quotient graph $\Gamma/N$ is defined as the graph for which the vertices are the orbits of acting $N$ on $V(\Gamma)$, and two vertices $A, B \in V(\Gamma/N)$ are adjacent if, and only if, there exists $a \in A$ and $b \in B$ such that $\{a, b\} \in E(\Gamma)$. The covering graph is regular or $N$-covering if there is a semiregular subgroup $N$ of the automorphism group $\text{Aut}(\overline{\Gamma})$ such that graph $\Gamma$ is isomorphic to the quotient graph $\dfrac{\overline{\Gamma}}{N}$. If $N$ is an elementary abelian (EA), then $\overline{\Gamma}$ is called an EA covering of $\Gamma$. Given a graph $\Gamma$ and a subgroup $G$ of $Aut(\Gamma)$, $\Gamma$ is *G*-vertex-transitive (GVT), *G*-edge-transitive (GET), or GAT if $G$ is transitive on the $V(\Gamma)$, $E(\Gamma)$, or $A(\Gamma)$, respectively. If $G = Aut(\Gamma)$ then $\Gamma$ is called a VT, ET or AT graph, respectively. A subspace $W$ of a vector space $V$ is said to be invariant subspace (IV) with respect to a linear transformation $T$ if $T(W) \subseteq W$. A matrix group (MG) is a group $G$ consisting of invertible matrices over a specified field $K$, with the operation of matrix multiplication.

A powerful and important tool topology and graph theory is covering techniques. Regular covering is an active and interesting topic in algebraic graph theory. Tutte in [1,2] studied finite cubic arc-transitives and showed that every arc-transitive graph of degree 3 has an order of the form $2np$, where $n > 0$ and a prime number $p$. Conder and

Dobcsányi [3,4] classified the trivalent *s*-regular graphs up to order 2048 with MAGMA software [5]. Cheng and Oxley classified the AT graphs of order $2p$ (see Table 1 in [6]). Talebi and Mehdipoor classified cubic semisymmetric graphs of order $18p^n$ in [7]. By using the covering technique, *s*-regular graphs with order $2p^2$, $2p^3$, $4p$, $4p^2$, $6p$, $6p^2$, $8p$, $8p^2$, $10p$, $12p$, $10p^2$, $14p$, $16p$, $28p$, $36p$, $44p$, $52p$, $66p$, $68p$, $76p$, $22p$, $22p^2$, $10p^3$, $3p^2$ and $6p^2$ were classified in [8–22]. Kosari et al. in [23] investigated new results in graphs. The automorphism lifting problem in the context of elementary abelian covers was studied by Malnič, Marušič and Potočnik [24]. Their results have been successfully applied in order to classify elementary abelian covers with specific symmetric properties for a number of small cubic or tetravalent graphs, namely, the complete graph $K_4$ [11], the $Q_3$ graph [12] the Heawood graph [24], the Petersen graph [25], the Möbius–Kantor graph [26] and the Octahedron graph [27]. Most recently, Talebi and Mehdipoor investigated semisymmetric $\mathbb{Z}_p$-covers of the *C*20 graph [28]. In this paper, we classify R-AT-EA-covers of the *C*13 graph, by using concepts of linear algebra.

Graph $\Gamma := C13$ is an AT tetravalent graph, which was defined in [29]. The order and size of this graph are 13 and 26, respectively. See Figure 1.

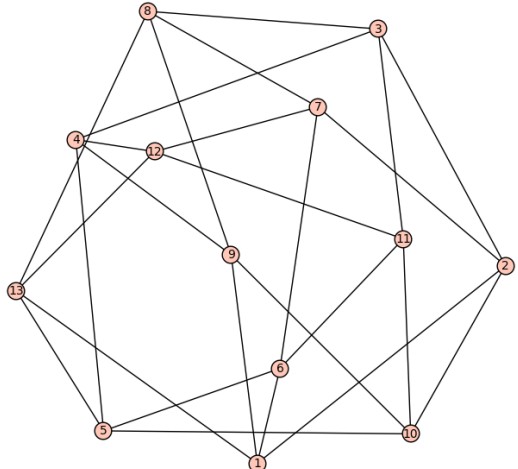

**Figure 1.** *C*13 graph.

$V(\Gamma) = \{1, 2, 3, 4, 5, 6, 7, 8, 9, 10, 11, 12, 13\}$
$E(\Gamma) = \{\{1, 2\}, \{1, 6\}, \{1, 9\}, \{1, 13\}, \{2, 3\}, \{2, 7\}, \{2, 10\}, \{3, 4\}, \{3, 8\},$
$\{3, 11\}, \{4, 5\}, \{4, 9\}, \{4, 12\}, \{5, 6\}, \{5, 10\}, \{5, 13\}, \{6, 7\}, \{6, 11\}, \{7, 8\},$
$\{7, 12\}, \{8, 9\}, \{8, 13\}, \{9, 10\}, \{10, 11\}, \{11, 12\}, \{12, 13\}\}.$

The automorphisms of *C*13 graph are

$\delta = (2, 9, 13, 6)(3, 4, 12, 11)(5, 7, 10, 8),$
$\sigma = (1, 2, 3, 4, 5, 6, 7, 8, 9, 10, 11, 12, 13).$

Then, $|Aut(C13)| = 52$. $Aut(C13)$ acts transitively on $V(C13)$, $E(C13)$ and $A(C13)$. We will see by using Sage software (S-S) [30] that $Aut(C13)$ has one AT subgroup $\langle \delta, \sigma^2 \rangle$.

Let $T$ be an ST of the *C*13 graph with edges

$$\{1, 2\}, \{1, 6\}, \{1, 9\}, \{1, 13\}, \{2, 3\}, \{2, 7\}, \{2, 10\}, \{3, 4\}, \{3, 8\},$$
$$\{3, 11\}, \{4, 5\}, \{4, 12\}.$$

By choosing $T$, we can consider in this graph a $T$-reduced VA, that is, the voltage values on the arcs of tree $T$ are the identity. The CT arcs are as follows:

$$\gamma_1 = (5, 6), \gamma_2 = (6, 7), \gamma_3 = (7, 8), \gamma_4 = (8, 9),$$
$$\gamma_5 = (9, 10), \gamma_6 = (10, 11), \gamma_7 = (11, 12), \gamma_8 = (12, 13), \gamma_9 = (13, 8),$$
$$\gamma_{10} = (13, 5), \gamma_{11} = (12, 7), \gamma_{12} = (11, 6), \gamma_{13} = (10, 5), \gamma_{14} = (9, 4).$$

Let $\Gamma$ be a graph and $N$ be a finite group. We denote the reverse as an arc $\omega$ with $\omega^{-1}$. A voltage assignment (VA) of $\Gamma$ is a function $\varphi : A(\Gamma) \to N$ such that $\varphi(\omega^{-1}) = \varphi(\omega)^{-1}$ for each arc $\omega \in A(\Gamma)$. The voltage is the value of $\varphi$, and $N$ is the voltage group. The graph $\Gamma \times_{\varphi} N$ ($Cov(\Gamma, \varphi)$) obtained from a VA $\varphi : A(\Gamma) \to N$ has vertex set $V(\Gamma) \times N$ and edge set $E(\Gamma) \times N$; then, an edge $(e, g)$ of $\Gamma \times N$ joins a vertex $(\omega, g)$ to $(b, \varphi(\omega)g)$ for $\omega = (a, b) \in A(\Gamma)$ and $g \in N$, where $e = \{a, b\}$. By considering the VA arcs, we can create a VA on walks [31], for example the voltage on a walk $W : \omega_1, \omega_2, \dots, \omega_n$, is $\varphi(\omega_1)\varphi(\omega_2) \dots \varphi(\omega_n)$. The derived graph $\Gamma \times_{\varphi} N$ is a covering of $\Gamma$. By defining $(a, g')^g = (a, g'g)$ for any $g \in N$ and $(a, g') \in V(\Gamma \times_{\varphi} N)$, is an $N$-covering, for any $a \in V(\Gamma)$. The reverse is also true. For a spanning tree (ST) $T$ of the graph $\Gamma$, a VA $\varphi$ is called $T$-reduced if the voltages on the tree arcs are the identity. If $T$ is an arbitrary fixed ST, then Gross and Tucker [32] proved that every regular covering $\overline{\Gamma}$ of a graph $\Gamma$ can be obtained from a $T$-reduced VA of $\overline{\Gamma}$. Assume that $\overline{\Gamma}$ is an $N$-covering of $\Gamma$. [33] If $\tau \in Aut(\Gamma)$ and $\overline{\tau} \in Aut(\overline{\Gamma})$ satisfy $\overline{\tau}p = p\tau$, where $p : \overline{\Gamma} \to \Gamma$, then $\overline{\tau}$ is a lift of $\tau$, and $\tau$ the projection of $\overline{\tau}$. A regular covering projection $p : \overline{\Gamma} \to \Gamma$ is VT, ET or AT if a VT, ET or AT subgroup of $Aut(\Gamma)$ lifts under $p$.

Let $\Gamma$ be a graph and $W_1, W_2$ be two walks of $\Gamma$. We show that the fundamental group of a graph $\Gamma$ is the set of all reduced walks equipped with the product $W_1 W_2$ by $\pi(\Gamma)$. The fundamental group of $\Gamma$ is called $\pi(\Gamma, v)$ at $v$. In general, the fundamental group is not a free group. Therefore, by abelianizing $\pi(\Gamma, v)$, the first homology group $HG_1(\Gamma)$ is obtained. It is not necessarily a free $\mathbb{Z}$-module. Suppose that $n_e + n_s$ is the minimal number of generators of $\pi(\Gamma, v)$, where $n_s$ is the number of semiedges and $n_e$ is the number of cotree (CT) loops and links relative to some ST, such that $HG_1(\Gamma) \cong \mathbb{Z}^{n_e} \times \mathbb{Z}_2^{n_s}$. [24] Observe that

$$HG_1(\Gamma, \mathbb{Z}_p) \cong \begin{cases} \mathbb{Z}_p^{n_e + n_s} & p = 2 \\ \mathbb{Z}_p^{n_e} & p \geq 3. \end{cases}$$

given a connected graph $\Gamma$ and a subgroup $G \leq Aut(\Gamma)$. Let $T$ be a ST of $\Gamma$ and a set of arcs $\{\gamma_1, \dots, \gamma_n\} \subseteq A(\Gamma)$ include precisely one arc from each edge in $E(\Gamma \setminus T)$. Suppose that $B_T$ is the corresponding basis of $HG_1(\Gamma, \mathbb{Z}_p)$ determined by $\{\gamma_1, \dots, \gamma_n\}$. Furthermore, denote by $G^{*h} = \{\tau^{*h} | \tau \in G\} \leq GL(HG_1(\Gamma, Z_p))$ the induced action of $G$ on $HG_1(\Gamma, Z_p)$, and let $M_G \leq \mathbb{Z}_p^{n \times n}$ be the matrix representation of $G^{*h}$ with respect to the basis $B_T$. The dual group including all transposes of matrices in $M_G$ is denoted by $M_G^t$.

Proposition 1 was obtained from [24], Proposition 6.3 and Corollary 6.5. This proposition is very important and widely used in the presentation of R-AT-EA-covers of the $C13$ graph.

**Proposition 1.** *Let $T$ be an ST of a connected graph $\Gamma$, and let the set $\{\gamma_1, \gamma_2, \dots, \gamma_r\} \subseteq A(\Gamma)$ include precisely one arc from each CT edge. Let $\varphi : A(\Gamma) \to Z_p^{d \times 1}$ be a VA on $\Gamma$ that is identical on $T$, and let $Z(\varphi) = [\varphi(\gamma_1), \varphi(\gamma_2), \dots, \varphi(\gamma_r)]^t$. Therefore, a group $G \leq Aut(\Gamma)$ lifts under $p_{\varphi} : Cov(\Gamma, \varphi) \to \Gamma$ if and only if the induced subspace $\langle Z(\varphi) \rangle$ is a d-dimensional(d-dime) $M_G^t$-IV.*

The main purpose for finding all regular EA coverings of a graph is finding all IVs of a MG. Now, we express the following theorem, which is basic for finding all IVs of MG.

**Theorem 1** ((Maschke's theorem)). *Let $V$ be a representation of the finite group $G$ over a field $F$ in which $|G|$ is invertible. Let $W$ be an invariant subspace of $V$. Then, there exists an invariant subspace $W_1$ of $V$ such that $V = W \oplus W_1$ as representations.*

## 2. R-AT-AE-Covers of the $C13$ Graph

In this section, we introduce all of the (connected) R-AT-EA-covers of the C13 graph with projection $p : \overline{\Gamma} \to C13$. Notation $C_{\gamma_i}$ is the fundamental closed walk, that is, $C_{\gamma_i}$ is one cycle of graph $\Gamma$ containing exactly an arc $\gamma_i$ of CT. Notation $B_T$ is the standard ordered basis of $HG1(C13, \mathbb{Z}_p)$ related to the ST $T$, and the arcs $\gamma_i (i = 1, \dots, 14)$, respectively. Here, we state Lemma 1.

**Lemma 1.** *Let A and B be the transposes of the matrices that indicate the linear transformations* $\delta^{*h}$, *and* $\sigma^{*h}$ *relative to* $B_T$. *Therefore*

$$
A = \begin{bmatrix}
0 & 0 & 0 & 0 & 0 & 0 & 0 & 0 & 0 & 0 & 1 & 0 & 0 & 1 \\
0 & 0 & 0 & 0 & -1 & 0 & 0 & 0 & 0 & 0 & 0 & 0 & 0 & 0 \\
0 & 0 & 0 & 0 & 1 & 0 & 0 & 0 & 0 & 0 & 0 & 0 & 1 & -1 \\
0 & 0 & 0 & 0 & 0 & 0 & 0 & 0 & 0 & -1 & 0 & 0 & 0 & 1 \\
0 & 0 & 0 & 1 & 0 & 0 & 0 & 0 & 1 & 0 & 0 & 0 & 0 & 0 \\
0 & 0 & 0 & -1 & 0 & 0 & 0 & 0 & 0 & 0 & 0 & 0 & 0 & -1 \\
0 & 0 & 0 & 0 & 0 & 0 & 1 & 0 & 0 & 0 & 0 & 0 & 0 & 0 \\
0 & 0 & 0 & 0 & 0 & 0 & -1 & 0 & 0 & 0 & 0 & 1 & 0 & 1 \\
-1 & 0 & 0 & 0 & 0 & 0 & 0 & 0 & 0 & 0 & 0 & 0 & 0 & -1 \\
0 & 1 & 0 & 0 & 0 & 0 & 0 & 0 & 0 & 0 & -1 & 0 & 0 & -1 \\
0 & 0 & 0 & 0 & -1 & -1 & -1 & 0 & 0 & 0 & 0 & 0 & 0 & 1 \\
0 & 0 & 0 & 0 & 0 & 0 & 0 & 0 & 0 & 0 & 0 & 0 & 0 & 1 \\
0 & 0 & -1 & -1 & 0 & 0 & 0 & 0 & 0 & 0 & -1 & 0 & 0 & -1 \\
0 & 0 & 0 & 0 & 0 & 0 & 0 & -1 & 0 & 0 & 0 & 0 & 0 & -1
\end{bmatrix}
$$

$$
B = \begin{bmatrix}
1 & 1 & 0 & 0 & 0 & 0 & 0 & 0 & 0 & 0 & 0 & 0 & 0 & 0 \\
0 & 0 & 1 & 0 & 0 & 0 & 0 & 0 & 0 & 0 & 0 & 0 & 0 & 0 \\
0 & 0 & 0 & 1 & 0 & 0 & 0 & 0 & 0 & 0 & 0 & 0 & 0 & 1 \\
0 & 0 & 0 & 0 & 1 & 0 & 0 & 0 & 0 & 0 & 0 & 0 & 0 & -1 \\
0 & 0 & 0 & 0 & 0 & 1 & 0 & 0 & 0 & 0 & 0 & 0 & 0 & 0 \\
0 & 0 & 0 & 0 & 0 & 0 & 1 & 0 & 0 & 0 & 0 & 0 & 0 & 0 \\
0 & 0 & 0 & 0 & 0 & 0 & 0 & 1 & 0 & 1 & 0 & 0 & 0 & 0 \\
0 & 0 & 0 & 0 & 0 & 0 & 0 & 0 & 0 & -1 & 0 & 0 & 0 & 0 \\
0 & 0 & 0 & 0 & 0 & 0 & 0 & 0 & 0 & 0 & 0 & 0 & 0 & 1 \\
-1 & 0 & 0 & 0 & 0 & 0 & 0 & 0 & 0 & 0 & 0 & 0 & 0 & 0 \\
0 & 0 & 0 & 0 & 0 & 0 & 0 & 0 & 1 & -1 & 0 & 0 & 0 & 0 \\
0 & 0 & 0 & 0 & 0 & 0 & 0 & 0 & 0 & 0 & 1 & 0 & 0 & 0 \\
-1 & 0 & 0 & 0 & 0 & 0 & 0 & 0 & 0 & 0 & 0 & 1 & 0 & 0 \\
0 & 0 & 0 & 0 & 0 & 0 & 0 & 0 & 0 & 0 & 0 & 0 & 1 & 0
\end{bmatrix}.
$$

**Proof.** By considering acting the automorphisms $\delta$ and $\sigma$ on $B_T$, we obtain the rows of these matrices. For instance, the permutation $\delta$ maps the cycle

$$[5, 6, 1, 2, 3, 4, 5]$$

corresponding to $\gamma_1$, to the cycle

$$[7, 2, 1, 9, 4, 12, 7].$$

Since the second cycle is the sum of the base cycles corresponding to $\gamma_{11}$ and $\gamma_{14}$, the first row of matrix $A$ is obtained. This is

$$(0, 0, 0, 0, 0, 0, 0, 0, 0, 1, 0, 0, 1).$$

Similar to the above, we can obtain the matrices $A$ and $B$.  □

By using S-S , $m_A(t) = t^4 - 1$ and $m_H(t) = t^{13} - 1$ are the minimal polynomials of $A$ and $H = B^2$. Assume that $p$ is a prime. $\xi$ is a primitive 13th root of unity in $\mathbb{Z}_p$. The minimal polynomial $m_H(x)$ is decomposed into irreducible factors over $\mathbb{Z}_p$. See the following statements:

$$
\begin{cases}
(t-1)^{13} & p = 13 \\
(t-1)(t-\xi)(t-\xi^2)(t-\xi^3)(t-\xi^4)(t-\xi^5)(t-\xi^6)(t-\xi^7)(t-\xi^8)(t-\xi^9)(t-\xi^{10})(t-\xi^{11})(t-\xi^{12}) & p \equiv 1 \bmod 13 \\
(t-1)((t-\xi)(t-\xi^3)(t-\xi^9))((t-\xi^2)(t-\xi^5)(t-\xi^6))((t-\xi^4)(t-\xi^{10})(t-\xi^{12}))((t-\xi^7)(t-\xi^8)(t-\xi^{11})) & p \equiv 3, 9 \bmod 13 \\
(t-1)((t-\xi)(t-\xi^3)(t-\xi^4)(t-\xi^9)(t-\xi^{10})(t-\xi^{12}))((t-\xi^2)(t-\xi^5)(t-\xi^6)(t-\xi^7)(t-\xi^8)(t-\xi^{11})) & p \equiv 4, 10 \bmod 13 \\
(t-1)((t-\xi)(t-\xi^5)(t-\xi^8)(t-\xi^{12}))((t-\xi^2)(t-\xi^3)(t-\xi^{10})(t-\xi^{11}))((t-\xi^3)(t-\xi^6)(t-\xi^7)(t-\xi^9)) & p \equiv 5, 8 \bmod 13 \\
(t-1)(t^{12} + t^{11} + t^{10} + t^9 + t^8 + t^7 + t^6 + t^5 + t^4 + t^3 + t^2 + t + 1) & p \equiv 2, 6, 7, 11 \bmod 13 \\
(t-1)((t-\xi)(t-\xi^{12}))((t-\xi^2)(t-\xi^{11}))((t-\xi^5)(t-\xi^8))((t-\xi^6)(t-\xi^7))((t-\xi^4)(t-\xi^9))((t-\xi^3)(t-\xi^{10})) & p \equiv -1 \bmod 13.
\end{cases}
$$

Now, it is sufficient to see $A$ and $H$ as matrices over the splitting field $\mathbb{Z}_p(\xi)$. To find $\langle B, H \rangle$-IVs over $\mathbb{Z}_p$, every IV over $\mathbb{Z}_p$ is a direct sum of minimal IVs over $\mathbb{Z}_p(\xi)$. By a straightway calculation and applying Lemma 1, we have

$$ker(H - \xi^2 I) = \langle v_1 \rangle,$$
$$ker(H - \xi^4 I) = \langle v_2 \rangle,$$
$$ker(H - \xi^6 I) = \langle v_3 \rangle,$$
$$ker(H - \xi^8 I) = \langle v_4 \rangle,$$
$$ker(H - \xi^{10} I) = \langle v_5 \rangle,$$
$$ker(H - \xi^{12} I) = \langle v_6 \rangle,$$
$$ker(H - \xi I) = \langle v_7 \rangle,$$
$$ker(H - \xi^3 I) = \langle v_8 \rangle,$$
$$ker(H - \xi^5 I) = \langle v_9 \rangle,$$
$$ker(H - \xi^7 I) = \langle v_{10} \rangle,$$
$$ker(H - \xi^9 I) = \langle v_{11} \rangle,$$
$$ker(H - \xi^{11} I) = \langle v_{12} \rangle,$$
$$ker(H - I) = \langle v_{13}, v_{14} \rangle$$

where

$$v_1 = \begin{bmatrix} \frac{\xi^{10}}{\xi^3+1} \\ \frac{(\xi-1)\xi^{10}}{\xi^2-\xi+1} \\ -\frac{1}{2}\frac{(7\xi^{12}-7\xi^{11}+4\xi^{10}-\xi^9+\xi-4)}{-\frac{1}{2}\xi^3-\frac{1}{2}\xi^{-3}-4\xi-4\xi^{-1}+2\xi^2+2\xi^{-2}+5} \\ -(\xi^2-\xi+1)^{-1} \\ -\frac{1}{2}\frac{(7\xi^{12}-4\xi^{11}+\xi^{10}-\xi^2+4\xi-7)}{-\frac{1}{2}\xi^3-\frac{1}{2}\xi^{-3}-4\xi-4\xi^{-1}+2\xi^2+2\xi^{-2}+5} \\ -\frac{\xi^2-1}{\xi(\xi^{10}+1)} \\ -\frac{(\xi-1)\xi^4}{\xi^2-\xi+1} \\ \frac{\xi^6}{\xi^3+1} \\ \xi^{11} \\ -\frac{\xi^{10}+1}{\xi^2(\xi^6+2\xi^3+1)} \\ -\frac{\xi^8+\xi^7+\xi^6+\xi+1}{\xi^6} \\ -\frac{\xi^8+\xi^7+\xi^2+\xi+1}{\xi^2} \\ \xi^2 \\ 1 \end{bmatrix}, \quad v_2 = \begin{bmatrix} \frac{1}{\xi^6(\xi^6+1)} \\ -\frac{(16\xi^{12}-15\xi^{11}+12\xi^{10}-8\xi^9+4\xi^8-\xi^7+\xi^5-4\xi^4+8\xi^3-12\xi^2+15\xi-16)\xi^6}{3\xi^{12}-8\xi^{11}+12\xi^{10}-16\xi^9+20\xi^8-24\xi^7+26\xi^6-24\xi^5+20\xi^4-16\xi^3-12\xi^2-8\xi+3} \\ -\frac{(4\xi^{12}-\xi^{11}+\xi^9-4\xi^8+8\xi^7-12\xi^6+15\xi^5-16\xi^4+16\xi^3-15\xi^2+12\xi-8)\xi^6}{3\xi^{12}-8\xi^{11}+12\xi^{10}-16\xi^9+20\xi^8-24\xi^7+26\xi^6-24\xi^5+20\xi^4-16\xi^3-12\xi^2-8\xi+3} \\ (\xi^4-\xi^2+1)^{-1} \\ -\frac{\xi^6(\xi^9-\xi^7+\xi^2-1)}{\xi^{10}-\xi^8+\xi^6+\xi^4-\xi^2+1} \\ -\frac{1}{\xi^7-1}(\xi^2-\xi^{-2}) \\ -\frac{(\xi^2-1)\xi^8}{\xi^4-\xi^2+1} \\ \frac{\xi^{12}}{\xi^6+1} \\ \xi^9 \\ -\frac{\xi^3}{\xi^6+1} \\ -\xi^4-\xi^3-\xi^2-\xi-1 \\ -\frac{\xi^4+\xi^3+\xi^2+\xi-1}{\xi^4} \\ \xi^4 \\ 1 \end{bmatrix},$$

$$v_3 = \begin{bmatrix} \frac{\xi^8(\xi^4+1)}{\xi^8+2\xi^4-1} \\ -\frac{\xi^{12}-2\xi^{11}+4\xi^{10}-2\xi^9+2\xi^8-4\xi^7+2\xi^6-\xi^5+2\xi^4-\xi^3+\xi-2}{\xi^{12}-\xi^{11}+2\xi^{10}-4\xi^9+2\xi^8-3\xi^7+6\xi^6-3\xi^5+2\xi^4-4\xi^3-2\xi^2-\xi+1} \\ -\frac{\xi^5(\xi^9-\xi^6-\xi^2+1)}{\xi^8+2\xi^4-1} \\ -(\xi^6-\xi^3+1)^{-1} \\ -\frac{(\xi^{10}+\xi^6-\xi^4-1)\xi^4}{\xi^8+2\xi^4-1} \\ -\frac{\xi^7(\xi^9-\xi^7-\xi^3+1)}{\xi^8+2\xi^4-1} \\ \frac{\xi^7(\xi^9-\xi^6-\xi^2+1)}{\xi^8+2\xi^4-1} \\ -\frac{1}{(\xi^4+1)\xi^4} \\ \xi^6 \\ -\frac{\xi^2}{\xi^4+1} \\ -\frac{\xi^{11}+\xi^8+\xi^5+\xi^3+1}{\xi^5} \\ -\frac{\xi^{11}+\xi^8+\xi^6+\xi^3+1}{\xi^6} \\ \xi^6 \\ 1 \end{bmatrix}, \quad v_4 = \begin{bmatrix} \frac{\xi^2}{1+\xi} \\ (\xi^7-\xi^6+\xi^5-\xi^4+\xi^3-\xi^2+\xi-1)\xi^2 \\ -\frac{\xi^5(\xi^5-1)}{1+\xi} \\ -\frac{\xi(\xi^4+1)}{1+\xi} \\ \frac{\xi^4-\xi^3+\xi^2-\xi+1}{\xi^4} \\ -\frac{(\xi^2+1)(\xi^5-\xi^4+\xi-1)}{\xi^4} \\ \frac{\xi^5-1}{\xi(1+\xi)} \\ \frac{1}{\xi(1+\xi)} \\ \xi^5 \\ -\frac{\xi^{12}+1}{\xi^5(\xi^2+2\xi+1)} \\ -\frac{\xi^{11}+\xi^9+\xi^7+\xi^5+1}{\xi^5} \\ -\frac{\xi^{11}+\xi^6+\xi^4-\xi^2+1}{\xi^6} \\ \xi^8 \\ 1 \end{bmatrix},$$

$$v_5 = \begin{bmatrix} \dfrac{1}{\xi^2(\xi^2+1)} \\[4pt] -\dfrac{(\xi^3-1)\xi^8}{\xi^2+1} \\[4pt] -\dfrac{(\xi^3-1)\xi^5}{\xi^2+1} \\[4pt] -\dfrac{(\xi^{11}+\xi^5+\xi^3+1)\xi^2}{\xi^4+2\xi^2+1} \\[4pt] \dfrac{(\xi^{11}-\xi^{10}-\xi^8+1)\xi^2}{\xi^4+2\xi^2+1} \\[4pt] \dfrac{(\xi^3-1)\xi^7}{\xi^2+1} \\[4pt] \dfrac{(\xi^3-1)\xi^4}{\xi^2+1} \\[4pt] \dfrac{\xi^4}{\xi^2+1} \\[4pt] \xi^3 \\[4pt] -\dfrac{\xi}{\xi^2+1} \\[4pt] -\dfrac{\xi^9+\xi^5+\xi^4+\xi-1}{\xi^4} \\[4pt] -\dfrac{\xi^9+\xi^8+\xi^5+\xi^4+1}{\xi^5} \\[4pt] \xi^{10} \\[4pt] 1 \end{bmatrix}, \quad v_6 = \begin{bmatrix} \dfrac{1}{\xi^5(\xi^5+1)} \\[4pt] -\dfrac{\xi-1}{\xi^6(\xi^5+1)} \\[4pt] -\dfrac{\xi-1}{\xi^7(\xi^5+1)} \\[4pt] -\dfrac{(\xi^8+\xi^6+\xi+1)\xi^5}{\xi^{10}+2\xi^5+1} \\[4pt] -\dfrac{(\xi^{12}-\xi^8+\xi^7-1)\xi^5}{\xi^{10}+2\xi^5+1} \\[4pt] -\dfrac{(19\xi^{12}-15\xi^{11}+8\xi^{10}-8\xi^8+15\xi^7-19\xi^6+20\xi^5-20\xi^4+20\xi^3-20\xi^2+20\xi-20)\xi^5}{\xi^{10}-4\xi^9+8\xi^8-12\xi^7+16\xi^6-18\xi^5+16\xi^4-12\xi^3+8\xi^2+4\xi+1} \\[4pt] \dfrac{(\xi-1)\xi^{10}}{\xi^5+1} \\[4pt] \dfrac{\xi^{10}}{\xi^5+1} \\[4pt] \xi \\[4pt] -\dfrac{\xi(\xi^7-\xi^6+\xi^5-\xi^4+\xi^3-\xi^2+\xi-1)}{\xi^9-\xi^8+\xi^7-\xi^6+\xi^5+\xi^4-\xi^3+\xi^2-\xi+1} \\[4pt] -\dfrac{\xi^{10}+\xi^7+\xi^4+\xi^3-1}{\xi^4} \\[4pt] -\dfrac{\xi^{10}+\xi^7+\xi^6+\xi^3+1}{\xi^6} \\[4pt] \xi^{12} \\[4pt] 1 \end{bmatrix},$$

$$v_7 = \begin{bmatrix} \dfrac{\xi^{10}}{\xi^5+1} \\[4pt] \dfrac{(\xi-1)\xi^{10}}{\xi^5+1} \\[4pt] \dfrac{\xi-1}{\xi^2(\xi^5+1)} \\[4pt] -\dfrac{\xi^5(\xi^6-\xi^5+\xi^4-\xi^3+\xi^2-\xi+1)}{\xi^4-\xi^3+\xi^2-\xi+1} \\[4pt] -\dfrac{\xi^5(\xi-1)}{\xi^5+1} \\[4pt] -\dfrac{(20\xi^{12}-20\xi^{11}+20\xi^{10}-20\xi^9+20\xi^8-19\xi^7+15\xi^6-8\xi^5+8\xi^3-15\xi^2+19\xi-20)\xi^5}{\xi^{10}-4\xi^9+8\xi^8-12\xi^7+16\xi^6-18\xi^5+16\xi^4-12\xi^3+8\xi^2+4\xi+1} \\[4pt] -\dfrac{(\xi-1)\xi^7}{\xi^5+1} \\[4pt] \dfrac{1}{\xi^5(\xi^5+1)} \\[4pt] \xi^{12} \\[4pt] -\dfrac{\xi(\xi^7-\xi^6+\xi^5-\xi^4+\xi^3-\xi^2+\xi-1)}{\xi^9-\xi^8+\xi^7-\xi^6+\xi^5+\xi^4-\xi^3+\xi^2-\xi+1} \\[4pt] -\dfrac{\xi^{10}+\xi^7+\xi^6+\xi^3+1}{\xi^6} \\[4pt] -\dfrac{\xi^{10}+\xi^7+\xi^4+\xi^3-1}{\xi^4} \\[4pt] \xi \\[4pt] 1 \end{bmatrix}, \quad v_8 = \begin{bmatrix} \dfrac{\xi^4}{\xi^2+1} \\[4pt] \dfrac{(\xi^3-1)\xi^4}{\xi^2+1} \\[4pt] \dfrac{(\xi^3-1)\xi^7}{\xi^2+1} \\[4pt] -\dfrac{\xi^2(\xi^8+1)}{\xi^2+1} \\[4pt] -\dfrac{\xi^2(\xi^3-1)}{\xi^2+1} \\[4pt] -\dfrac{(\xi^3-1)\xi^5}{\xi^2+1} \\[4pt] -\dfrac{(\xi^3-1)\xi^8}{\xi^2+1} \\[4pt] \dfrac{1}{\xi^2(\xi^2+1)} \\[4pt] \xi^{10} \\[4pt] -\dfrac{\xi}{\xi^2+1} \\[4pt] -\dfrac{\xi^9+\xi^8+\xi^5+\xi^4+1}{\xi^5} \\[4pt] -\dfrac{\xi^9+\xi^5+\xi^4+\xi-1}{\xi^4} \\[4pt] \xi^3 \\[4pt] 1 \end{bmatrix},$$

$$v_9 = \begin{bmatrix} \dfrac{1}{(1+\xi)\xi} \\[4pt] -(\xi^7-\xi^6+\xi^5-\xi^4+\xi^3-\xi^2+\xi-1)\xi^4 \\[4pt] \dfrac{(\xi^5-1)\xi^4}{1+\xi} \\[4pt] -\dfrac{(\xi^{12}+\xi^9+\xi^8+1)\xi}{\xi^2+2\xi+1} \\[4pt] -\dfrac{(\xi^4-1)\xi^9}{\xi^9-\xi^5-1} \\[4pt] \dfrac{(\xi^2+1)(\xi^5-\xi^4+\xi-1)}{\xi^3} \\[4pt] -\dfrac{(\xi^4-1)\xi^6}{\xi^9-\xi^5-1} \\[4pt] \dfrac{\xi^2}{1+\xi} \\[4pt] \xi^8 \\[4pt] -\dfrac{\xi^{12}+1}{\xi^5(\xi^2+2\xi+1)} \\[4pt] -\dfrac{\xi^{11}+\xi^6+\xi^4-\xi^2+1}{\xi^6} \\[4pt] -\dfrac{\xi^{11}+\xi^9+\xi^7+\xi^5+1}{\xi^5} \\[4pt] \xi^5 \\[4pt] 1 \end{bmatrix}, \quad v_{10} = \begin{bmatrix} \dfrac{\xi^9(\xi^4+1)}{\xi^8+2\xi^4-1} \\[4pt] \dfrac{2\xi^{12}-\xi^{11}+\xi^9-2\xi^8+\xi^7-2\xi^6+4\xi^5-2\xi^4+2\xi^3+4\xi^2+2\xi-1}{\xi^{12}-\xi^{11}+2\xi^{10}-4\xi^9+2\xi^8-3\xi^7+6\xi^6-3\xi^5+2\xi^4-4\xi^3-2\xi^2-\xi+1} \\[4pt] -\dfrac{\xi^7(\xi^9-\xi^7-\xi^3+1)}{\xi^8+2\xi^4-1} \\[4pt] -\dfrac{\xi^6}{\xi^6-\xi^3+1} \\[4pt] \dfrac{(\xi^9-\xi^7-\xi^3+1)\xi^4}{\xi^8+2\xi^4-1} \\[4pt] -\dfrac{\xi^5(\xi^9-\xi^6-\xi^2+1)}{\xi^8+2\xi^4-1} \\[4pt] \dfrac{\xi^5(\xi^9-\xi^7-\xi^3+1)}{\xi^8+2\xi^4-1} \\[4pt] \dfrac{\xi^8}{\xi^4-1} \\[4pt] \xi^6 \\[4pt] -\dfrac{\xi^2}{\xi^4+1} \\[4pt] -\dfrac{\xi^{11}+\xi^8+\xi^6+\xi^3+1}{\xi^6} \\[4pt] -\dfrac{\xi^{11}+\xi^8+\xi^5+\xi^3+1}{\xi^5} \\[4pt] \xi^7 \\[4pt] 1 \end{bmatrix},$$

$$v_{11} = \begin{bmatrix} \frac{\xi^{12}}{\xi^6+1} \\ -\frac{\left(15\xi^{12}-12\xi^{11}+8\xi^{10}-4\xi^9+\xi^8-\xi^6+4\xi^5-8\xi^4+12\xi^3-15\xi^2+16\xi-16\right)\xi^6}{3\xi^{12}-8\xi^{11}+12\xi^{10}-16\xi^9+20\xi^8-24\xi^7+26\xi^6-24\xi^5+20\xi^4-16\xi^3-12\xi^2-8\xi+3} \\ \frac{\left(12\xi^{12}-15\xi^{11}+16\xi^{10}-16\xi^9+15\xi^8-12\xi^7+8\xi^6-4\xi^5+\xi^4-\xi^2+4\xi-8\right)\xi^6}{3\xi^{12}-8\xi^{11}+12\xi^{10}-16\xi^9+20\xi^8-24\xi^7+26\xi^6-24\xi^5+20\xi^4-16\xi^3-12\xi^2-8\xi+3} \\ -\frac{\xi^4}{\xi^4-\xi^2+1} \\ -\frac{\left(\xi^9-\xi^6+\xi^2-1\right)\xi^6}{\xi^{12}+2\xi^6+1} \\ \frac{\xi^2-1}{\left(\xi^4-\xi^2+1\right)\xi^2} \\ \frac{\left(\xi^2-1\right)\xi^7}{\xi^4-\xi^2+1} \\ \frac{1}{\xi^6\left(\xi^6+1\right)} \\ \xi^4 \\ -\frac{\xi^3}{\xi^6+1} \\ -\frac{\xi^4+\xi^3+\xi^2+\xi-1}{\xi^4} \\ -\xi^4-\xi^3-\xi^2-\xi-1 \\ \xi^9 \\ 1 \end{bmatrix}, \quad v_{12} = \begin{bmatrix} \frac{\xi^6}{\xi^3+1} \\ -\frac{(\xi-1)\xi^4}{\xi^2-\xi+1} \\ -\frac{1}{2}(\xi^{12}-\xi^4+4\xi^3-7\xi^2+7\xi-4) \\ \frac{-\frac{1}{2}(\xi^{12}-\xi^4+4\xi^3-7\xi^2+7\xi-4)}{\left(-\frac{1}{2}\xi^3-\frac{1}{2}\xi^{-3}-4\xi-4\xi^{-1}+2\xi^2+2\xi^{-2}+5\right)} \\ -\frac{\xi^2}{\xi^2-\xi+1} \\ \frac{-\frac{1}{2}(4\xi^{12}-\xi^{11}+\xi^3-4\xi^2+7\xi-7)}{\left(-\frac{1}{2}\xi^3-\frac{1}{2}\xi^{-3}-4\xi-4\xi^{-1}+2\xi^2+2\xi^{-2}+5\right)} \\ \frac{\xi-1}{\left(\xi^2-\xi+1\right)\xi} \\ \frac{(\xi-1)\xi^{10}}{\xi^2-\xi+1} \\ \frac{1}{\xi^3\left(\xi^3+1\right)} \\ \xi^2 \\ -\frac{\xi^{10}+1}{\xi^2\left(\xi^6+2\xi^3+1\right)} \\ -\frac{\xi^8+\xi^7+\xi^2+\xi+1}{\xi^2} \\ -\frac{\xi^8+\xi^7+\xi^6+\xi+1}{\xi^6} \\ \xi^{11} \\ 1 \end{bmatrix},$$

$$v_{13} = \begin{bmatrix} -1 \\ 0 \\ 0 \\ -1 \\ 0 \\ 0 \\ 0 \\ -1 \\ 1 \\ 1 \\ 0 \\ 0 \\ 1 \\ 1 \end{bmatrix}, \quad v_{14} = \begin{bmatrix} 1 \\ 0 \\ 0 \\ 0 \\ 0 \\ 0 \\ 1 \\ 0 \\ -1 \\ 1 \\ 1 \\ 0 \\ 0 \end{bmatrix}.$$

By the linear transformation $A$, we calculate the images of $v_i$, $1 \le i \le 14$ as follows:

$$Av_1 = -\left(\frac{\xi^6}{\xi^3+1}+1\right)v_8,$$

$$Av_2 = -\left(\frac{\xi^{12}}{\xi^6+1}\right)v_3,$$

$$Av_3 = -\left(\frac{\xi^9\left(\xi^4+1\right)}{\xi^8+2\xi^4-1}+1\right)v_{11},$$

$$Av_4 = -\left(\frac{1}{(1+\xi)\xi}+1\right)v_6,$$

$$Av_5 = -\left(\frac{\xi^4}{\xi^2+1}+1\right)v_1,$$

$$Av_6 = -\left(\frac{\xi^{10}}{\xi^5+1}+1\right)v_9,$$

$$Av_7 = -\left(\frac{1}{\xi^5\left(\xi^5+1\right)}+1\right)v_4,$$

$$Av_8 = -\left(\frac{1}{\xi^2\left(\xi^2+1\right)}+1\right)v_{12},$$

$$Av_9 = -\left(\frac{\xi^2}{1+\xi}+1\right)v_7,$$

$$Av_{10} = -\left(\frac{\xi^8\left(\xi^4+1\right)}{\xi^8+2\xi^4-1}+1\right)v_2,$$

$$Av_{11} = -\left(\frac{1}{\xi^6\left(\xi^6+1\right)}+1\right)u_{10},$$

$$Av_{12} = -\left(\frac{\xi^{10}}{\xi^3+1}+1\right)v_5,$$

$$Av_{13} = v_{14},$$

$$Av_{14} = v_{13}.$$

Let $V$ be a minimal $\langle A, H \rangle$-IV. Let

$$V_1 = \langle v_{13}, v_{14} \rangle$$

and

$$V_2 = \langle v_1, v_2, v_3, v_4, v_5, v_6, v_7, v_8, v_9, v_{10}, v_{11}, v_{12} \rangle$$

over the field $\mathbb{Z}_p(\xi)$. Suppose first that $V \cap V_2 = 0$. Then $V \le V_1$. More exactly, $V \le \langle v_{13}, v_{14} \rangle \cap A\langle v_{13}, v_{14} \rangle$. Since $Av_{13} = v_{14}$ and $Av_{14} = v_{13}$, we have $V \le \langle v_{13}, v_{14} \rangle$. All one-dime subspaces of $\langle v_{13}, v_{14} \rangle$ are $\langle v_{13} \rangle$ and $\langle sv_{13} + v_{14} \rangle$, where $s \in \mathbb{Z}_P$. If $v_{13} \in V$, then $Av_{13} = v_{14} \in V$ and hence $V = W_1 := \langle v_{13}, v_{14} \rangle$. Let $sv_{13} + v_{14} \in V$. Then, $A(sv_{13} + v_{14}) = sv_{14} + v_{13} \in V$. Since $\langle sv_{13} + v_{14}, sv_{14} + v_{13} \rangle = \langle v_{13}, v_{14} \rangle$, we have $V = W_1$.

Now, assume that $V$ intersects $V_2$ nontrivially, such that $V$ must include one of the minimal $H$-IVs in $V_2$ that are 1-dime subspaces of the spaces $\langle v_1 \rangle$, $\langle v_2 \rangle$, $\langle v_3 \rangle$, $\langle v_4 \rangle$, $\langle v_6 \rangle$, $\langle v_5 \rangle$, $\langle v_8 \rangle$, $\langle v_7 \rangle$, $\langle v_9 \rangle$, $\langle v_{10} \rangle$, $\langle v_{12} \rangle$ and $\langle v_{11} \rangle$. We examine the following cases.

**Case I:** $v_1 \in V$.

Suppose that $v_1 \in V$. Then, $Av_1 \in V$, implying that $v_8 \in V$ and hence $Av_8 \in V$. Since $Av_8 = -(\frac{1}{\xi^2(\xi^2+1)} + 1)v_{12}$, we have $v_{12} \in V$. By considering $Av_{12}$, one can see that $V = W_2 := \langle v_1, v_{12}, v_5, v_8 \rangle$.

**Case II:** $v_2 \in V$.

Suppose that $v_2 \in V$. Then $Av_2 = -(\frac{\xi^{12}}{\xi^6+1})v_3$ and hence $v_3 \in V$. By noting that $Av_3 = -(\frac{\xi^9(\xi^4+1)}{\xi^8+2\xi^4-1} + 1)v_{11}$, we have $v_{11} \in V$. Now by considering $Av_{11}$, we can see $v_{10} \in V$. Therefore, $V = W_3 := \langle v_2, v_3, v_{10}, v_{11} \rangle$.

**Case III:** $v_3 \in V$.

Assume that $v_3 \in V$. Then, $Av_3 \in V$, implying that $v_{11} \in V$ and hence by considering $Av_{11}$, we have $v_{10} \in V$. Finally, by noting that $Av_{10} = -(\frac{\xi^8(\xi^4+1)}{\xi^8+2\xi^4-1} + 1)v_2$, one can see that $V = W_3$.

**Case IV:** $v_4 \in V$.

Let $v_4 \in V$. Then, $Av_4 = -(\frac{1}{(1+\xi)\xi} + 1)v_6$. By considering $Av_6$, we have $v_9 \in V$. Finally, by noting that $Av_9 = -(\frac{\xi^2}{1+\xi} + 1)v_7$ and $Av_7 = -(\frac{1}{\xi^5(\xi^5+1)} + 1)v_4$, one can see that $V = W_4 := \langle v_4, v_6, v_9, v_7 \rangle$.

In the remaining cases, by considering $Av_i$, $5 \leq i \leq 14$, we have one of the top IVs.

Now, we use Maschke's theorem to find all IVs. See the following lemma.

**Lemma 2.** *All proper nontrivial $\langle A, H \rangle$-IVs on the splitting field $\mathbb{Z}_p(\xi)$ are*
$W_1 = \langle v_{13}, v_{14} \rangle$,
$W_2 = \langle v_1, v_5, v_8, v_{12} \rangle$,
$W_3 = \langle v_2, v_{10}, v_3, v_{11} \rangle$,
$W_4 = \langle v_4, v_7, v_6, v_9 \rangle$,
$W_5 := \langle v_{13}, v_{14}, v_1, v_5, v_8, v_{12} \rangle$,
$W_6 := \langle v_{13}, v_{14}, v_2, v_3, v_{10}, v_{11} \rangle$,
$W_7 := \langle v_{13}, v_{14}, v_6, v_4, v_9, v_7 \rangle$,
$W_8 := \langle v_1, v_5, v_8, v_{12}, v_2, v_3, v_{10}, v_{11} \rangle$,
$W_9 := \langle v_1, v_5, v_8, v_{12}, v_4, v_7, v_6, v_9 \rangle$,
$W_{10} := \langle v_4, v_7, v_6, v_9, v_2, v_3, v_{10}, v_{11} \rangle$,
$W_{11} := \langle v_{13}, v_{14}, v_1, v_5, v_8, v_{12}, v_6, v_4, v_7, v_9 \rangle$,
$W_{12} := \langle v_{13}, v_{14}, v_6, v_4, v_9, v_7, v_2, v_{10}, v_3, v_{11} \rangle$,
$W_{13} := \langle v_{13}, v_{14}, v_1, v_5, v_8, v_{12}, v_2, v_{10}, v_3, v_{11} \rangle$,
$W_{14} := \langle v_1, v_5, v_8, v_{12}, v_4, v_7, v_6, v_9, v_2, v_3, v_{10}, v_{11} \rangle$.

Eventually, we delete element $\xi$ from bases for the spaces $W_i$, $2 \leq i \leq 14$. We first show that the subspaces $W_2$ are not $\langle A, H \rangle$-IVs over $\mathbb{Z}_p$ where $p \not\equiv 1 \bmod 13$. Suppose that for some $a_i, b_i, c_i, d_i \in \mathbb{Z}_p(\xi)$, $(a_0, a_1, \ldots, a_{11}, b_0, b_1, \ldots, b_{11}, c_0, c_1, \ldots, c_{11}, d_0, d_1, \ldots, d_{11}) \neq (0, 0, \ldots, 0, 0, 0, \ldots, 0, 0, 0, \ldots, 0, 0, 0, \ldots, 0)$, $\sum_{i=0}^{11} a_i \xi^i v_1 + b_i \xi^i v_{12} + c_i \xi^i v_5 + d_i \xi^i v_8 \in \mathbb{Z}_p^{14}$. By considering the coordinates, we see there are not $a_i, b_i, c_i, d_i \in \mathbb{Z}_p$ such that $\sum_{i=0}^{11} a_i \xi^i v_1 + b_i \xi^i v_{12} + c_i \xi^i v_5 + d_i \xi^i v_8 \in \mathbb{Z}_p^{14}$. It can be seen by using the same method that the remaining spaces $W_i$, $3 \leq i \leq 14$ are not $\langle A, H \rangle$-IVs over $\mathbb{Z}_p$.

According to the above explanation, the following theorem is obtained.

**Theorem 2.** *Let $p$ be a prime. Suppose that $\overline{\Gamma}$ is an AT cover of the C13 graph, and the group $G$ lifts. In Figure 2, the connected R-AT-EA-p-cover of the C13 graph is introduced:*

| inv.sub | $\varphi(\gamma_1)$ | $\varphi(\gamma_2)$ | $\varphi(\gamma_3)$ | $\varphi(\gamma_4)$ | $\varphi(\gamma_5)$ | $\varphi(\gamma_6)$ | $\varphi(\gamma_7)$ |
|---|---|---|---|---|---|---|---|
| $\langle W_1 \rangle$ | $\begin{array}{c} -1 \\ 1 \end{array}$ | $\begin{array}{c} 0 \\ 0 \end{array}$ | $\begin{array}{c} 0 \\ 0 \end{array}$ | $\begin{array}{c} -1 \\ 0 \end{array}$ | $\begin{array}{c} 0 \\ 0 \end{array}$ | $\begin{array}{c} 0 \\ 0 \end{array}$ | $\begin{array}{c} 0 \\ 0 \end{array}$ |
| inv.sub | $\varphi(\gamma_8)$ | $\varphi(\gamma_9)$ | $\varphi(\gamma_{10})$ | $\varphi(\gamma_{11})$ | $\varphi(\gamma_{12})$ | $\varphi(\gamma_{13})$ | $\varphi(\gamma_{14})$ |
| $\langle W_1 \rangle$ | $\begin{array}{c} -1 \\ 1 \end{array}$ | $\begin{array}{c} 1 \\ 0 \end{array}$ | $\begin{array}{c} 1 \\ -1 \end{array}$ | $\begin{array}{c} 0 \\ 1 \end{array}$ | $\begin{array}{c} 0 \\ 1 \end{array}$ | $\begin{array}{c} 1 \\ 0 \end{array}$ | $\begin{array}{c} 1 \\ 0 \end{array}$ |

**Figure 2.** R-AT- AE-p-cover of the C13 graph.

We cannot use Maschke's theorem for $p = 2, 13$ because the hypothesis does not hold. Then, we use S-S to complete R-AT-AE-p-covers of the *C*13 graph.

## 3. Conclusions

Symmetric graphs are used in computer networks, so studying these graphs is very important. For this reason, researchers in algebraic graph theory have been seriously classifying these graphs and studying their properties since around the year 2000. With a linear representation of automorphisms acting on the first homology group of the graph, the EA covering projections of a graph can be found. Essentially, the main purpose of this method is finding IVs of MGs over prime fields. It should be noted that applying S-S is very effective in presenting the main result. In this paper, we classified R-AT-EA-covers of the *C*13 graph. In Table 1, this cover, along with the VS on the arcs of CT, was introduced. We plan to study R-AT- AE-covers of quintic graphs in the future. Our next purpose is to investigate ET, semisymmetric and half-arc-transitive coverings of tetravalent graphs.

**Author Contributions:** Methodology, Q.X., A.K., A.A.T. and N.M.; validation, Q.X. and N.M.; formal analysis, A.A.T. and Q.X.; investigation, A.K. and A.A.T.; data curation, N.M., Q.X., A.K. and A.A.T.; writing—original draft preparation, A.A.T. and N.M.; writing—review and editing, Q.X., A.K., A.A.T. and N.M.; visualization, N.M. and A.K.; supervision, A.K. and Q.X.; project administration, Q.X. and A.A.T.; funding acquisition, Q.X., A.K. and A.A.T. All authors have read and agreed to the published version of the manuscript.

**Funding:** This research received no external funding.

**Institutional Review Board Statement:** Not applicable.

**Informed Consent Statement:** Not applicable.

**Data Availability Statement:** Not applicable.

**Conflicts of Interest:** The authors declare no conflict of interest.

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
