# Peer review of "Classification of Arc-Transitive Elementary Abelian Covers of the C13 Graph"

_symmetry, doi:10.3390/sym14051066_

Round 1

Reviewer 1 Report

In this paper the authors classified regular AT abelian covers of the C13 graph, where C13 is a tetravalent graph of order 13 defined in [25] and AT means that the group Aut(C13)  is transitive on the set of arcs the graph.

In the study are used different softwares, but the theoretical part must be improved by defining correctly all the notions or by giving all necessary references. The paper it is carelessly written. There are elementary wrong statements and other errors. For example, I notice:

- p. 1, line 21: “in V(Γ/N)” must be replaced with “A,B in V(Γ/N)”; 

- p. 2, lines 43 and 68: “show” must be replaced with “denote”; 

          line 44: “A(X)” must be replaced with “A(Γ)”; 

          line 46: “V(X)” must be replaced with “V(Γ)” (in fact the lines 43-51 contain more unclear parts); 

           line 72: “proposition 1, proposition 6.3, corollary 6.5” must be replaced with “Proposition 1, Proposition 6.3, Corollary 6.5” (these type of errors appear more times in the paper e.g. lines 114, 139, 233 and so on); 

- p. 3, lines 81-83: The statement of Maschke’s theorem must be corrected;

          line 99: “ba” must be replaced with “be”; 

- p. 5, line 130: “by” must be replaced with “By”;

- p. 8, lines 206, 213: W4 are defined in two different form;

- p. 9, line 236: the statement is not correct,

 - voltage assignment and a primitive root is denoted by same ξ, and so on.

Author Response

  “Reply to Referee 1”

We appreciate the constructive suggestions given by the referee 1 and incorporate them in the revised version which are given as follows:

  • in this article, we used Sage software for calculations  
  • symmetric graphs are used in computer networks, so studying these graphs is very important.
  • line 21:“in V(Γ/N)” replaced with “A, in V(Γ/N)”.
  • we define correctly all the notions in article. 
  • lines 43 and 68:“show replaced with “denote”. 
  • line 44:A(X)” replaced with “A(Γ)”.
  • line 46:V(X)” replaced with “V(Γ)”
  • line 72:“proposition, corollary, lemma, table” replaced with “Proposition, Corollary, Lemma, Table.
  • lines 81-83: the statement of Maschke’s theorem replaced with (Maschke's theorem) Let V be a representation of the finite group G over a field F in which |G| is invertible. Let W be an invariant subspace of V. Then there exists an invariant subspace of V such that V = W W1 as representations. Theorem 1.2.1(A Course in Finite Group Representation Theory, Peter Webb).
  • line 99:“ba” replaced with “be”; 
  • line 130:“by” replaced with “By”;
  • lines 206, 213: we correct W4.
  • line 236:the statement is replaced with “We cannot use the Maschke's theorem because the hypotheses for p = 2, 13 does not holds.”
  • voltage assignment denoted by and a primitive root is denoted by ξ.

Thank you so much from Referee 1 for his/her Useful and constructive comments that helped us to improve quality of paper!

Reviewer 2 Report

In the first read, it gives the impression that the authors put everything into a box and throw it down from a hill. The manuscript seems to be dealing with a nice problem but the authors, trying to make it more interesting, have lost their aim and orientation. Even the title is long and wrong. It could be "Classification of Arc-transitive Elementary Abelian Covers of C13 Graph". 

What is a C13 graph and what is its importance? The authors define C13 in Section 2 but it is mentioned in the title, abstract and introduction. As, according to the figure, it is not one of the classical graphs in graph theory, the authors should mention  its details much earlier, before the first use in the text. Also there are too many important notions such as homology group, fundamental group, etc from algebraic topology, but I could not see how important are these notions for this manuscript. I have not checked by a programme, but I have enough reasons to suspect that one of the two automorphisms, the one of order 4, may not be an automorphism, looking at the cycles. There are many similar problems with the design and order of the manuscript. I suggest a serious major revision. It needs reordering and a lot of explanations of the important parts. Also it needs to be shortened by omitting or reducing the size of the parts giving not-so-closely related notions.

Author Response

  “Reply to Referee 2”

We appreciate the constructive suggestions given by the referee 2 and incorporate them in the revised version which are given as follows:

  • title replaced with "Classification of Arc-transitive Elementary Abelian Covers of C13 Graph".
  • symmetric graphs are used in computer networks, so studying these graphs is very important.
  • symmetric and semisymmetric graphs are widely used in computer networks. These graphs were introduced about 1940. Especially in recent years, the study of the properties of these graphs has been considered by researchers in algebraic graph theory.  The researchers directly studied the graphs with small orders that did not involve long and complicated calculations of automorphisms and their action on vertices and edges. However, due to the increasing order of graphs and complicated calculations of automorphisms, they introduced these graphs with software. For example, we can refer to the classification of cubic symmetric graphs up to 2048 vertices. [Trivalent (cubic) symmetric graphs on up to 2048 vertices, J (2006).

 http://www.math.auckland.ac.nz conder/symmcubic2048list.txt.] 

In high orders, we need the structure of small graphs (Base graphs).  For example, in classification cubic symmetric graphs of order 4p, we consider cubic symmetric graph of order 4 as base graph ().  If the structure of this graph is defined then we can introduce a new graph X 

 (called the voltage graph) from this graph. Graph X has order 4p.  Now, by using concept lift a automorphism, we complete the classification of graphs of order 4p. The C13 graph (Circulant cayley graph) was introduced by Potocnik and Wilson, http://jan.ucc.nau.edu/ swilson/C4Site/index.html, This graph and all its structures are on this website and can be used as an atlas. Well-known graphs are mostly cubic graphs.  We can use these graphs to classify the other cubic graphs. To classify tetravalent graphs, the base graphs must be regular of valency 4. Therefore, the structure of tetravalent graphs is very important and significant. For this purpose, we can use from this website (http://jan.ucc.nau.edu/ swilson/C4Site/index.html,) to study the structure of these graphs and classify graphs from higher orders. To classify symmetric graphs, considering that these graphs must have a symmetric subgroup, we have considered the C13 graph.

  • definition of C13 graph and its details were moved to the introduction section.
  • notions such as homology group, fundamental groupare used in description before Proposition 1, which is the most important proposition in classification arc-transitive cover of X in this paper. Let X  be  a connected graph and   Suppose that T be a spanning tree  and ,  belong to arc set X (A(X)) include precisely one arc from each edge in  .  Assume that   be the corresponding basis of HG1(X, ) determined by ,  Further, denote by  = { |  G} ≤ GL(HG1(X, ))( Homology group) the induced action of G on HG1(X, Zp)( Homology group), and let  ≤  be the matrix representation of  with respect to the basis BT.  The dual group including of all transposes of matrices in  is denoted by  .
  • in general, the method of this article is as follows:
  • in the first step, we firstdetermine spanning tree of the base graph X and consider the complement tree (cotree or CT).
  • in the second step, we need to find the automorphism group of this graph (Aut(X)).
  • in the third step, we need to get the symmetric subgroups of graph X. The researchersused Magma software to calculate symmetric subgroups. The orders to calculate symmetric subgroups were not written in the help of software, directly.  Help of this software only checked for symmetric graph  where (G = Aut(X)). Therefore, to investigate symmetric subgroups, we wrote a program to find these subgroups by using sage software.
  • in the fourth step, we have to write these subgroups according to the generators of the automorphism group.
  • in the fifth step, the action of symmetric subgroup generators on the fundamental cycles should be seen and then the result should be rewritten according to the fundamental cycles. See matrices A and B.
  • in the sixth step, we decompose the polynomial matrices on the splitter field. We used an algorithm in sage software.
  • in the seventh step, we calculated the kernel values and eigenvectors and the action of matrices on these eigenvectors.To do this, we used sage software. However, for the action of matrices and linear combinations of eigenvectors, we implemented the Ax = b solution algorithm.
  • in the eighth step, we introduced 14 invariant subspaces on the splitter field by using the Maschke’s theorem.
  • in the ninth step, we found the invariant subspaces on .
  • in the tenth step, we introduced regular arc-transitive elementary abelian covers (R-AT-EA-covers) of the C13 graph in Table 1.
  • finally, In the eleventh step, use the sage software where the hypotheses of  Maschke’s theoremdoes not holds.
  • the automorphism group of this graph was calculated with Sage software.(Sage web site http://www.sagemath.org)

  • we tried to resize the vectors and matrices as much as possible according to the journal text file.

Thank you so much from Referee 2 for his/her Useful and constructive comments that helped us to improve quality of paper!

Reviewer 3 Report

Introduction should be presented in a section only.
- The state-of-the-art literature is not presented in this work, the authors need to show what has been done in the literature, and what authors did in this work. I suggest authors to create a new section as
a literature review or related work to show the literature history.
- Several keywords are not defined well; authors need to check all keywords once again to provide the abbreviation of all keywords rather than what the authors presented in the submission.
- The results should be compared with other existing methods.
- The research method of this study is not clear.
- Sensitivity analysis is not presented.
- The conclusion must be extended and improved based on study limitations and recommendations for further work.

Author Response

  “Reply to Referee 3”

We appreciate the constructive suggestions given by the referee 3 and incorporate them in the revised version which are given as follows:

  • introduction present in a section only.
  • in the introduction we have added referencesto show the work history and importance of this issue.
  • all keywords were defined in the text of the article.
  • in general, the method of this article is as follows:

  • in the first step, we firstdetermine spanning tree of the base graph X and consider the complement tree (cotree or CT).
  • in the second step, we need to find the automorphism group of this graph (Aut(X)).
  • in the third step, we need to get the symmetric subgroups of graph X. The researchersused Magma software to calculate symmetric subgroups. The orders to calculate symmetric subgroups were not written in the help of software, directly.  Help of this software only checked for symmetric  graph  where (G = Aut(X)). Therefore, to investigate symmetric subgroups, we wrote a program to find these subgroups by using sage software.
  • in the fourth step, we have to write these subgroups according to the generators of the automorphism group.
  • in the fifth step, the action of symmetric subgroup generators on the fundamental cycles should be seen and then the result should be rewritten according to the fundamental cycles. See matrices A and B.
  • in the sixth step, we decompose the polynomial matrices on the splitter field. We used an algorithm in sage software.
  • in the seventh step, we calculated the kernel values and eigenvectors and the action of matrices on these eigenvectors.To do this, we used sage software. However, for the action of matrices and linear combinations of eigenvectors, we implemented the Ax = b solution algorithm.
  • in the eighth step, we introduced 14 invariant subspaces on the splitter field by using the Maschke’s theorem.
  • in the ninth step, we found the invariant subspaces on .
  • in the tenth step, we introduced regular arc-transitive elementary abelian covers (R-AT-EA-covers) of the C13 graph in Table 1.
  • finally, In the eleventh step, use the sage software where the hypotheses of  Maschke’s theoremdoes not holds.
  • in conclusion, we explained what we want to do in the future.
  • the only method for studying arc transitive covers of graphs using the concepts of linear algebra is mentioned in this paper. The method of study is the same in all articles, but with the change of basic graphs, automorphism, symmetric subgroups, and everything changes. Hence, a new problem is presented. Note that the process of all articles is to provide the conditions for Proposition 1. Symmetric graphs are used in computer networks, so studying these graphs is very important. For this reason, researchers in algebraic graph theory have been seriously classifying these graphs and studying their properties since about 2000.

Thank you so much from Referee 3 for his/her Useful and constructive comments that helped us to improve quality of paper!

Round 2

Reviewer 1 Report

Now the paper is better than the previous.

Reviewer 2 Report

The authors made the required corrections and additions quite satisfactorily. There are still some minor corrections like the missing "is" in the first line of the Abstract. I suggest the authors to check the grammer throughly. 

Also I do not understand why the authors color-coded the paragraphs where they had no change. But it is not a problem. This recommendation is for the future reference to forthcoming works of the authors.

Reviewer 3 Report

It can be accepted now.